# The Hypothalamic–Pituitary–Thyroid Axis Equivalent in Normal and Cancerous Oral Tissues: A Scoping Review

**DOI:** 10.3390/ijms232214096

**Published:** 2022-11-15

**Authors:** Lisa Wu, Stephen Xu, Brian Yang, Jenny Yang, Claire Yee, Nicola Cirillo

**Affiliations:** Melbourne Dental School, The University of Melbourne, 720 Swanston Street, Carlton, VIC 3053, Australia

**Keywords:** head and neck cancer, oral squamous cell carcinoma, thyroid hormones, thyroid hormone receptors, thyrotropin-releasing hormone, thyroid-stimulating hormone

## Abstract

The hypothalamic–pituitary–thyroid (HPT) axis is crucial in regulating thyroid hormone levels that contribute to the development and homeostasis of the human body. Current literature supports the presence of a local HPT axis equivalent within keratinocytes of the skin, with thyroid hormones playing a potential role in cancer progression. However, this remains to be seen within oral tissue cells. An electronic search of Scopus and PubMed/Medline databases was conducted to identify all original publications that reported data on the production or effects of HPT axis components in normal or malignant cells of the oral cavity. The search identified 221 studies, of which 14 were eligible. Eight studies were retrospective analyses of clinical samples, one study involved both in vivo and in vitro experiments, and the remaining five studies were conducted in vitro using cell lines. The search identified evidence of effects of HPT components on oral cancer cells. However, there were limited data for the production of HPT axis components by oral tissues. We conclude that a possible role of the local HPT axis equivalent in the oral mucosa may not be established at present. The gaps in knowledge identified in this scoping review, particularly regarding the production of HPT components by oral tissues, warrant further investigation.

## 1. Introduction

The hypothalamic–pituitary–thyroid (HPT) axis is primarily responsible for maintaining normal circulating levels of thyroid hormones (TH)—thyroxine (T4) and triiodothyronine (T3)—essential to the development and homeostasis of all body tissue [1]. TH themselves exert negative feedback effects on their upstream HPT axis components, creating a complex negative neuroendocrine feedback system [2]. However, advances in research have shown that synthesis and direct target effects of neuroendocrine machinery like the HPT axis are not restricted to their primary system. For example, human skin has its own local neuroendocrine systems equivalent to the HPT axis, hypothalamic–pituitary–adrenal (HPA) axis, and more [3]. Human skin has been shown to express genes for HPT axis components, including thyroid-stimulating hormone (TSH) and TH-regulating molecules [4]. Furthermore, expression of thyrotropin-releasing hormone (TRH) and TSH receptors occurs within skin keratinocytes, melanocytes, fibroblasts, and hair follicles, enabling skin to be a direct target of thyroid hormone effects locally and systemically [5,6,7]. These HPT hormone effects on skin keratinocytes include proliferation, differentiation, and regulation of skin immunity and inflammation [7,8]. In hair follicles, changes in TH and TRH lead to stimulation of hair follicle proliferation, inhibition of apoptosis, and pigmentation [6,9].

Within the recent literature, TH are increasingly acknowledged for their role in cancer. Previous rodent models have demonstrated TH promoted growth and metastasis of tumor transplants, while epidemiological studies suggest that hyperthyroidism increases the risk of some solid malignancies [10,11]. In skin, loss of TH receptors is a common feature in some tumors and can increase aggressiveness of skin tumors [12]. Thus, the increasing evidence that other organs like the skin have local neuroendocrine systems not only deepens the understanding of human pathophysiology but also presents an opportunity for further research and potential therapeutics.

While keratinocytes are common to both skin and oral mucosa, their behavior and morphology differ with different gene expression profiles and response to injury [13,14,15]. The presence and effects of a local HPT axis within the oral mucosa have not been well documented [16], and hence there is currently a lack of detailed published literature available on this topic, which hinders the development of a systematic review question addressing the role of a putative oral HPT axis. A more exploratory type of investigation to assess the breadth of current literature is required and, as such, a scoping review was conducted to methodically analyze published data and to identify existing gaps in knowledge that can serve as the basis for further investigation. Hence, the aims of this scoping review were to document (i) evidence of the synthesis of HPT components by oral tissues and (ii) evidence for the effects of HPT components on normal and malignant oral tissues.

## 2. Materials and Methods

### 2.1. Protocol and Search Strategy

The protocol and search strategy for this scoping review was conducted in compliance with the Preferred Reporting Items for Systematic Reviews and Meta-Analysis (PRISMA) guidelines [17]. The search was completed within PubMed and Scopus databases during May 2022 with the following search string: (TRH OR TSH OR “thyrotropin-releasing hormone” OR “thyroid-stimulating hormone” OR Triiodothyronine OR Thyroxine OR “thyroid hormone receptor” OR THR OR THRB OR THRA OR TRHR OR “TRH receptor” OR “TSH receptor”) AND (“oral keratin*” OR “oral fibro*” OR OSCC OR oral squamous cell carcinoma OR“ oral cancer” OR “oral carcinoma” OR “head and neck ca*” OR “oral muco*” OR OPMD OR “tongue squamous cell carcinoma” OR “oral potentially malignant dis*”).

### 2.2. Eligibility Criteria

As this scoping review aimed to systematically capture all evidence of production or effects of HPT components in oral tissues, the following inclusion criteria were used: (a) original study designs, (b) written in English, and (c) data reporting the production of HPT components or their effects in oral tissue. No time restriction was set. Exclusion criteria consisted of (a) nonoriginal articles (e.g., reviews, systematic reviews, guidelines), (b) written in languages other than English, and (c) no full text available. The HPT components considered in this review were TH, THR, TRH, TRHR, TSH, TSHR, and associated genes.

### 2.3. Study Selection and Date of Collection

The search results were recorded before automatic filtering of reviews and systematic reviews with duplicates removed. Study selection was divided into two phases. Following calibration of the reviewers, the first phase consisted of title and abstract screening split between three pairs—L.W. and B.Y., L.W. and J.Y, and L.W. and S.X.—under the supervision of the senior investigator (N.C.). Cohen’s kappa for this stage of screening was 0.70, 0.78, and 0.93 respectively, showing moderate interrater reliability (Appendix A). Full-text articles were obtained for all results that met the inclusion criteria or that could not be definitively excluded. Full-text analysis against inclusion and exclusion criteria was then completed by two pairs of independent reviewers—J.Y. and S.X. and B.Y. and C.Y. Reasons for exclusion were recorded and any disagreement resolved by a third independent reviewer, L.W., and further reviewed by the senior investigator (N.C.). Data extraction was performed using a customized Excel extraction sheet.

## 3. Results

### 3.1. Overview of Search Results

The initial search of PubMed and Scopus yielded 221 results. Twenty-four studies were removed based on automatic filters for non-English articles, reviews, and systematic reviews, and 26 duplicates removed through manual screening. A further 66 publications were removed through title and abstract screening, leaving 105 for full-text retrieval and review. Critical analysis of full-text articles resulted in exclusion of 81 articles (specific reasons provided in Figure 1), with most articles excluded as they lacked original results on the production or effects of HPT axis hormones or receptors in oral tissue.

The final 14 studies were then eligible for inclusion in this scoping review. A basic overview of parameters of these eligible publications is discussed here. Study characteristics of the 14 studies [18,19,20,21,22,23,24,25,26,27,28,29,30,31], including the type of study, population, intervention, control, outcomes (PICO), biomarkers, assay methodology, and specimen types, are presented in Table 1.

### 3.2. Summary of Findings

#### 3.2.1. Loss of Heterozygosity (LOH) in THRB Gene Loci

Loss of heterozygosity involving the THRB gene loci in human samples was reported in four studies [19,25,26,29] summarized in Table 2. El-Naggar et al. [19] reported that three of nine informative primary head and neck squamous carcinoma specimens exhibited LOH at the THRB locus, with all three of these carcinomas showing DNA aneuploidy. Miyashita et al. [25] reported two of seven informative OSCC tissue samples exhibiting LOH on 3p24.1-22 (THRB) loci. Patridge et al. [26] found that 10 of 36 informative primary OSCC samples exhibited LOH at the 3p24-26 chromosomal region (which includes THRB loci). Rowley et al. [29] reported that 2 of 11 informative squamous cell carcinoma of the head and neck specimens exhibited LOH at the 3p24 chromosomal position. 

Patridge et al. [26] additionally reported that allelic imbalance (AI) at one or more loci involving the THRB gene loci was associated with reduced survival (HR = 4.21) (*p*-value = 0.0002). AI at 3p24-26 was also found to predict poor prognosis independently of other loci (HR = 3.93) (*p* = 0.002), and that AI at the 3p24-26 chromosomal region was found to be a better predictor of outcome than TNM staging.

#### 3.2.2. Methylated CpG Site of Thyrotropin-Releasing Hormone Gene Sequence

Methylation of CpG site in the TRH gene sequence in human samples was reported in one study [27]. Through bioinformatics, CpG site cg01009664 in the TRH gene sequence was found to have the highest difference in methylation level between healthy and cancerous cells. Pyrosequencing of microdissected samples revealed five CpG sites surrounding cg01009664, with the average methylation percentage in cancerous cells (52.96% ± 5.36%) being significantly higher than that in healthy cells (5.7% ± 0.85%) (*p* < 0.001).

Real time PCR at cg01009664 in oral rinse and swab samples was used as a screening marker in two cohorts. In both cohorts, the average TRH methylation levels of oral rinse from SCC subjects were significantly higher than that of healthy controls. Comparisons of other combinations of samples are reported in Appendix A. There were no significant differences in TRH methylation levels between sexes, ages, stages, or grades in any cancer samples.

Using receiver-operating characteristic analysis and a TRH methylation cutoff value of 3.31 ng/μL, the detection of OSCC from oral rinse had 86.15% sensitivity, 89.66% specificity, and 0.93 area under curve (AUC). With the same cutoff, the detection of oropharyngeal SCC from oral rinse had 82.61% sensitivity, 92.59% specificity, 0.93 AUC. For detecting OSCC, oral swab samples (91.30% sensitivity, 84.85% specificity, 0.97 AUC) were superior to oral rinse samples.

#### 3.2.3. THRA in Tongue Squamous Cell Carcinoma (TSCC) Progression

Isolation of 7SK chromatin by RNA purification (ChIRP)-Seq data and bioinformatic analysis was performed to identify interactions with 5’ regulatory regions and 5 motifs were obtained. Of the 5 motifs, motifs 3, 4, and 5 had similar binding motifs to PAX5, THRA, and FOXJ3 [31]. RT-qPCR revealed that FOXJ3 and THRA was found abundantly in SCC15 cells compared to PAX5. The results showed that 21 out of 27 genes had either knockdown of FOXJ3 or THRA. Amid these genes, there were 9 genes in common and 12 genes oppositely regulated by 7SK and FOXJ3/THRA. In particular, CXCL1, SYDE1, COL5A1, and HIF1A were identified to be negatively regulated by 7SK and positively regulated by FOXJ3 and THRA.

#### 3.2.4. TSH and Antithyroid Antibody Expression in OLP Patients

Two studies [28,30] looked at the expression of thyroid-stimulating hormone (TSH) and antithyroid antibodies in oral lichen planus (OLP) patients. Vehviläinen et al. [30] obtained negative results from both qRT-PCR and ddPCR for TSH from OLP lesions from both patients with hypothyroidism and without. However, Robledo-Sierra et al. [28] found that more patients in the OLP+/LT4+ group had low levels of FT3 (*p* = 0.0.387), while more patients from the OLP-/LT4+ group had high levels of FT4 (*p* = 0.142). There were no significant differences in the TSH levels (*p* = 0.5773), and no correlation between increased levels of antithyroid antibodies and clinical types of OLP lesions. The levels of TgAb (*p* = 0.2450), TPOAb (*p* = 0.1366) andTRAb (*p* = 1.0000) were similar in OLP+/LT4+ and OLP-/LT4+ groups [28]. Immunohistochemical analyses showed positive staining for TSHR in all OLP+/LT4+ patients in the basal layer of epithelium, while there were negative results for healthy controls [28]. The qPCR analysis presented higher expression of TSHR in patients than in healthy controls (*p* = 0.0008). The expression of Tg and TPO was not different between the groups [28].

#### 3.2.5. Effects of HPT Axis Components on Oral Cell Lines

Four papers [18,21,23,24] in the review investigated the effects of NDAT, resveratrol, thyroxine (T4) and STAT3 inhibitors on OSCC cell lines, namely, OECM-1 and SCC-25 cells (Appendix A). However, for the scope of this review, we were only interested in the effects of T4 on the two cell lines. In these studies, mRNA expression levels of PD-L1, CCND1, COX-2, BAD, BTLA, TNF-β1 and IL-1β were assessed. The concentration of T4 administered was 10^−7^ M and mRNA expression was measured using qPCR. Results of T4 treated cell cultures were compared to controls.

All studies showed that T4 induces PD-L1 expression. Proliferative genes (BTLA and CCND1) were shown to increase in the presence of T4. The proapoptotic gene BAD was shown to decrease in the presence of T4. T4 also upregulated TNF-β1 and IL-1β expression, and downregulated COX-2 expression.

One study [24] explored the effects of T4 at other concentrations. For OEC-M1 cells, relative mRNA expression and protein abundance increased in a dose-dependent manner. Compared to controls, T4 induced a 1.9-fold increase in PD-L1 expression (*n* = 3, *p* < 0.001) and a 1.5-fold increase in protein abundance (*n* = 3, *p* < 0.01). In SCC-25 cells, maximal mRNA expression was induced by 10^−7^ M concentration of T4, which induced a 2.7-fold increase in PD-L1 expression (*n* = 3, *p* < 0.001) and a 1.8-fold increase in protein abundance (*n* = 3, *p* < 0.001) compared to controls. For both cell lines, T4 induced pPI3K proteins that increased in a dose-dependent manner in SCC-25 cells and exhibited maximal protein abundance, with 10^−7^ M of T4 in OEC-M1 cells. This was also found in another study (18), where T4 showed an increase in protein abundance for pSer-STAT3, pPI3K and pERK 1/2 (*n* = 4, *p* < 0.05 for each protein) in OEC-M1 cells.

Formanek et al. [20] investigated the effects of various media additives on growth of human oral keratinocytes, including hydrocortisone, epidermal growth factor, insulin, bovine pituitary extract, transferrin, cholera toxin, adenine, and triiodothyronine, in varying concentrations and combinations, with cell proliferation measured by ^3^H-labeled thymidine incorporation. Results were expressed as a percentage of an additive-free control. Bovine pituitary extract and triiodothyronine, both showed slightly stimulatory effects only at their lowest concentrations (2 µg/mL and 10^−9^ M respectively). In combination additive experiments containing triiodothyronine, the highest proliferation was found in combination with insulin and hydrocortisone, with 166.5% at 48 h.

#### 3.2.6. TSH Levels and Association with Disease Outcome in Head and Neck Squamous Cell Carcinoma (HNSCC)

Jank et al. [22] reported on the association of TSH and CRYM (µ-crystallin) with disease outcome in HNSCC patients in a retrospective observational cohort study. At a 5-year follow-up, patients with low preoperative TSH levels had worse overall survival (OS) rates than patients with normal TSH (20% vs. 58%, *p* = 0.035). The study found an association between low TSH and OS (HR 2.99, *p* = 0.047), but not with disease-free survival (DFS).

Using antibody staining in a tissue microarray, it was found that CRYM+ was associated with better OS than CRYM- in a 5-year follow-up (78.6% vs. 56%, *p* = 0.027). No statistically significant improvement in DFS was found. No correlations were found between TSH and CRYM levels (*p* = 0.289). The study was not able to investigate whether high CRYM could abrogate effects of low TSH, as no CRYM+ TSH- patients were in the cohort.

## 4. Discussion

The results of this scoping review suggest that the limited evidence available in the scientific literature points toward a possible effect of HPT components on the physiology of normal and malignant oral keratinocytes. It is less clear, however, whether oral cells can themselves synthesize thyroid hormones and other neuroendocrine molecules of the HPT axis, such as TSH and TRH.

LOH involving THRB gene loci was reported to varying levels, which could suggest a role of THRB abnormalities in OSCC. However, as numerous studies use different biomarkers with different levels of informativeness, this conclusion cannot be readily drawn. Nevertheless, the evidence does suggest that LOH involving the THRB gene has potential prognostic significance. With these studies, techniques used could potentially produce confounding results through contamination of healthy cells in carcinoma samples and the number of PCR cycles, with more cycles increasing the chance of homodimers forming heterodimers [19]. Reaction failure [25] and samples being “not done” [26] were also reported.

Methylation of CpG sites in the promoter TRH gene in OSCC was demonstrated. Additionally, TRH methylation showed high sensitivity and specificity for OSCC and oropharyngeal SCC using oral rinse and oral swab, with oral swab having a higher sensitivity, presumably due to the collection of more cancerous epithelial cells. However, oral rinse may be more suitable for screening a larger population. Methylation status of TRH CpG sites has the potential to serve as a biomarker, considering the invasive nature of biopsy. False-positive results from oral swab were more likely to arise when the sample could only be taken from necrotic tissue over the lesion [27].

THRA potentially enhances tumor migration. One study looked at 7SK and its functional involvement in THRA and FOXJ3 in 7SK-mediated TSCC progression. THRA was found to be able to bind to T3, though the levels of expression of THRA remained unchanged in patients with TSCC [31]. Dysregulation of THRA could be involved in cancer, as mutations in the ligand-binding domain and zinc finger domain of THRA were observed in HNSCC patients, though there was no change in level of expression [31].

Two articles reviewed the expression of TSH and antithyroid antibodies in OLP patients. It is hypothesized that thyroid disease could result in the development of OLP. This is attributed to the similar molecular mimicry seen in both hypothyroidism and OLP, where there could be similar antigens and human autoantigens involved [30]. Vehviläinen et al. [30] reported that there were studies on antithyroid antibodies contributing to the initiation of an autoimmune response in the oral cavity. However, Robledo-Sierra et al. [28] found that there was no association between the levels of antithyroid antibodies and OLP. Patients with OLP without previously diagnosed thyroid disease did not have increased TgAb and TPOAb, as only 10% of the patients in the OLP+/LT4− group presented with increased serum levels. There were, however, elevated levels of TSH and reduced levels of FT4 associated with the presence of OLP [28]. The greater expression of TSH levels in OLP lesions meant that there is an immunological link between autoimmune thyroid disease as an etiology in OLP [28].

T4 is seen inducing expression of proliferation genes, such as PD-L1 and CCND1, and downregulates proapoptotic genes, such as BAD. Furthermore, T4 is seen modulating inflammatory genes, such as TNF-β1 and IL-1β, and downregulating COX-2, potentially altering cancer progression. Hence, Lin et al. [23] suggested that T4 contributes to the proliferation of cancer cells by increasing expression of checkpoint genes. In all these studies, the primary aim was not to identify effects of T4, but rather to investigate the ability of NDAT or resveratrol to reverse cell proliferation induced by T4, and thus these papers may provide more insight into possible therapeutic treatment of oral cancer cells than mechanism of cancer progression.

Hyperthyroidism has been shown to be associated with poorer survival rates in HNSCC patients [22]. These results are complementary to past studies that have shown hypothyroidism to be associated with better survival rate [32]. The study also shows CRYM levels could give some indication of disease outcome, with CRYM+ associated with better survival. The study highlights the importance of monitoring HPT components in HNSCC patients, indicating interventions to treat hyperthyroidism or induce hypothyroidism could have a positive impact on disease outcome. It also indicates areas for future investigation, such as interactions between HPV oncoproteins and thyroid hormones. The study had limited cohorts of interest and could not investigate the correlation between CRYM+ and low TSH, as no patient fit this category. The authors also concede a possibility of selection bias due to the retrospective nature of the study, as well as possible heterogeneous distribution of CRYM, as they used tissue microarray samples rather than whole tissue sections. The study design attempted to offset this by taking three random cores per specimen.

Certain additives in vitro can stimulate proliferation of oral keratinocytes [20]. Notably for this review, triiodothyronine and bovine pituitary extract as single additives showed slightly stimulatory effects at their lowest doses. Moderate proliferation was also exhibited in combinations containing triiodothyronine and bovine pituitary extract, especially when combined with insulin and hydrocortisone. However, this study does not explicitly address HPT interactions with HNSCC, and investigations into relevant hormones and their effects on oral cell proliferation could also highlight areas of interest in cancerous growth. The study is met with some limitations however, as not all results are consistent with past literature, and the authors acknowledge future research will be needed for long-term culture of oral cells.

We acknowledge that our scoping review has some limitations. Exclusion of non-English articles and articles that could not be retrieved may contain information that would otherwise have added to our knowledge on this topic. The use of two search engines for peer-reviewed articles and the selection of a specific search string without gray literature and manual search may have failed to identify relevant studies. In addition, full texts of eight articles could not be retrieved. Finally, the present report yielded only 14 articles, making it challenging to draw meaningful conclusions, especially for studies with minimal data and small samples.

Overall, the results of the present study show that there is scope for investigating whether the oral mucosa can act as a local HPT axis equivalent.

## 5. Conclusions

Our review revealed evidence of the effects of HPT axis hormones on malignant and non-malignant oral cells and alterations in HPT axis hormone and receptor genes in human oral squamous cell carcinoma, in addition to their association with disease outcome. However, no studies were identified that reported on the production of HPT axis components by oral tissues.

## Figures and Tables

**Figure 1 ijms-23-14096-f001:**
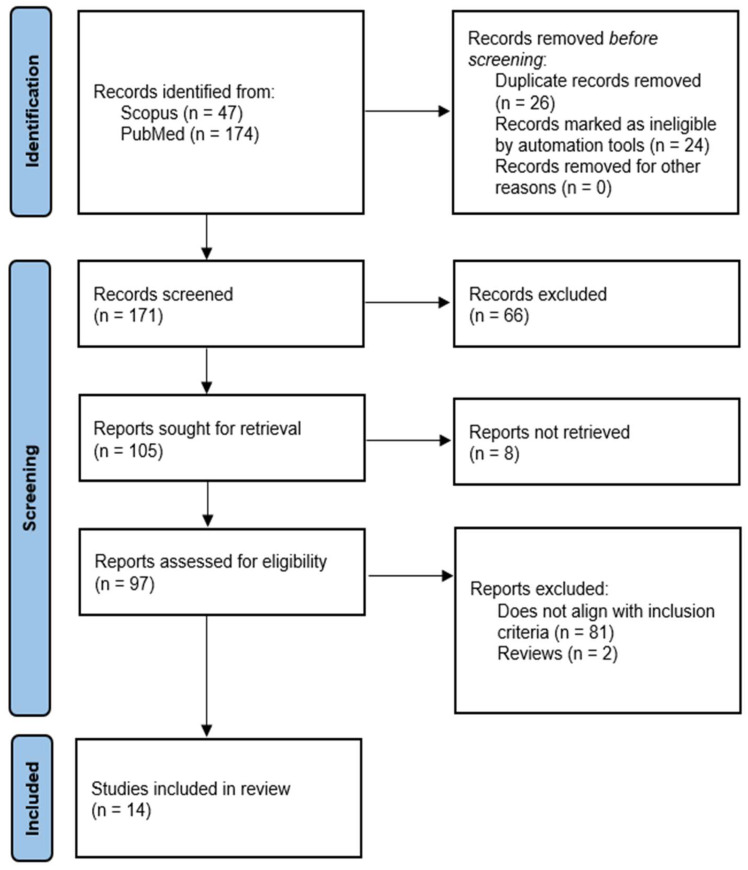
Flowchart for study selection in accordance with Preferred Reporting Items for Systematic Reviews and Meta-Analyses (PRISMA) guidelines.

**Table 1 ijms-23-14096-t001:** Characteristics of included studies.

Reference	Type of Study	Population/Specimen	Intervention	Control	Outcome	Biomarkers	Assay Methodology
Chen et al. (2019) [18]	in vitro	Human (OEC-M1, SCC-25)	10^−7^ M T4, resveratrol (40 μM or 10 μM) for 96 h or 24 h	Untreated cells	Proliferation and gene expression changes	IL-1b, CCND1, PD-L1 (CD274), COX-2, BAD, TGF-b1, STAT3, PI3K, ERK1/2	Cell proliferation assay, qPCR, confocal microscopy, WB analysis
el-Naggar et al. (1993) [19]	clinical	Human (HNSCC tissue specimens)	NA	Normal mucosa from same patients	LOH	Restriction enzymes at the D3F1552, D3S32, THRB loci	PCR, flow cytometry, southern blotting
Formanek et al. (1996) [20]	in vitro	Human (primary cultures of oral cells)	Hydrocortisone, insulin, EGF, T3, bovine pituitary extract, transferrin, cholera toxin, adenine, KGM	DMEM (additive free control)	Proliferation	3H-labeled thymidine	Proliferation assays using 3H-labeled thymidine incorporation. Flow cytometry
Ho et al. (2020) [21]	in vitro	Human (OEC-M1, SCC-25)	10^−7^ M T4, 40 μM resveratrol, 10^−7^ NDAT, STAT3 inhibitor (S31-201, 40 μM)	Unstimulated cells	Gene expression changes, cell viability	IL-1B, TNF-a, BAD, CCND1, PD-L1, COX-2.	Cell viability test (alamar Blue Assay kit), qPCR, confocal microscopy
Jank et al. (2021) [22]	clinical	Human HNSCC patients	NA	NR	Clinical out-come, TSH level, CRYM	µ-Crystallin. Thyrotropin	Tissue Microarray/IHC
Lin et al. (2019) [23]	in vitro	Human (OEC-M1, SCC-25)	10^−7^ M T4, 40 μM resveratrol and their combination for 24 h	Unstimulated cells	Gene expression, nuclear PD-L1	BAD, CCND1, PD-L1, BTLA	qPCR, confocal microscopic analysis
Lin et al. (2018) [24]	in vitro	Human (OEC-M1, SCC-25)	10^−8^ M to 10^−6^ M T4, 10^−8^ M to 10^−6^ M NDAT	Unstimulated cells	Expression/localization of PD-L1	PD-L1, pPI3K	qRT-PCR, Western blotting
Miyashita et al. (2008) [25]	clinical	Human OSCC tissue	NA	Adjacent non-neoplastic tissue	LOH and MSI	PTHR1, THRB, p53, APC, BRCA1, BRCA2, DCC, FHIT	H&E staining, laser captured microdissection, PCR, electrophoresis
Patridge et al. (1999) [26]	clinical	Human OSCC, blood	NA	Normal samples	LOH, AI, FAL	Microsatellite markers (3p, 8p21-23, 9p13-21 9q22, 13q14.2) p53, Rb, DCC	Toluidine blue staining, microdissection, PCR-RFLP, gel electrophoresis
Puttipanyalears et al. (2018) [27]	clinical	Human oral SCC (tissue, swab, rinse)	NA	Healthy human oral tissue	Methylation of CpG	cg01009664 TRH (CpG methylated site)	Microdissection, pyrosequencing, RT-PCR
Robledo-Sierra et al. (2018) [28]	clinical	Human blood samples from OLP patients	NA	Healthy subjects	Antithyroid antibodies and OLP (association)	Circulating TgAb, TPOAb, FT3, FT4 Abs; thyroid proteins (Tg, TPO, TSHR)	IHC and qPCR analyses
Rowley (1996) [29]	clinical	Human tumors of the head and neck	NA	Normal tissue samples from same patients	LOH, microsatellite instability	Microsatellite markers D3S1304D3S656, D3S1252D3S1293, THRB, and D3S1266.	PCR
Vehviläinen et al. (2020) [30]	clinical	Human oral mucosal tissue (OLP)	NA	Healthy controls without oral lesions	Measurement of TSH and TSHR expression	TSH and TSHR	qRT-PCR, ddPCR, IHC, H&E staining
Zhang et al. (2021) [31]	in vitro	Human (SCC15)	Transfection of cells by ShRNAs targeting 7SK	PLKO negative control	Expression of 7SK, cell proliferation and migration, apoptosis	NR	RT-qPCR, RNA sequencing, cell migration assay, cell proliferation assay, cytometry
	in vivo	BALB/c nude mice	Subcutaneous injection of 7SK-/- SCC15 cells at the armpits	Subcutaneous injection of control cells	Identification of enriched genes	NR	H&E staining, ChIRP-Seq analysis

Abbreviations: AI, allelic imbalance; ChiRP-Seq, chromatin isolation by RNA purification; EGF, epidermal growth factor; FAL, fractional allelic loss; H&E, hematoxylin and eosin; IHC, immunohistochemistry; LOH, loss of heterozygosity; MIS, microsatellite instability; NA, not applicable; NR, not reported; RT-PCR, reverse-transcription polymerase chain reaction.

**Table 2 ijms-23-14096-t002:** LOH in THRB gene loci.

Author (Year)	Human Carcinoma Sample	Chromosomal Location	LOH/Informative Sample (%)
el-Naggar et al. (1993) [19]	HNSCC	3p24	3/9 (33%)
Miyashita et al. (2008) [25]	OSCC	3p24.1-22	2/7 (28.6%)
Patridge et al. (1999) [26]	OSCC	3p24-26	10/36 (28%)
Rowley et al. (1996) [29]	Head and neck tumours	3p24	2/11 (18%)

## Data Availability

Full data sets are available upon reasonable request to the corresponding author.

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
