# Peer review of "The Hypothalamic–Pituitary–Thyroid Axis Equivalent in Normal and Cancerous Oral Tissues: A Scoping Review"

_ijms, 2022, doi:10.3390/ijms232214096_

Round 1

Reviewer 1 Report

The manuscript by Wu and colleagues describes a scoping review of the literature on the influence of the hypothalamic-pituitary-thyroid (HPT) axis on normal and cancerous oral tissue. Pubmed and Scopus databases were searched for articles investigating the effect of thyroid hormone and its receptor in oral tissues or cell lines. Fourteen studies were identified thus. Data from the studies were presented by loss of heterozygosity in thyroid hormone receptor gene, thyroid hormone receptor RNA expression, epigenetic modification of thyrotropin releasing hormone gene, antibody levels against thyroid stimulating hormone, exogenous thyroid hormone stimulations, correlation between thyroid stimulating hormone and disease outcome, and proliferation following thyroid hormone stimulation. From the results, the authors conclude that there is validity in investigating the expression of HPT components by oral mucosa cells.

General comments:

The overall premise of the review seems odd to me. The others clearly identify a gap of knowledge without scoping the literature (line 52f). Would the best way to answer this not be a study examining the effect of HPT component production by oral tissue rather than examining the limited literature that will not allow you to make adequate conclusions?

Some of the summarized results are based on a single study (section 2.2.2; 2.2.3; 2.2.6; and 2.2.7). Section 2.2.5 cites four papers, but all are from the same group. This diminishes their impact dramatically and may not warrant its own results section.

The authors seem to make no distinction if the results they are summarizing are obtained from patient samples, primary cells, or are from cell lines. There are also several studies that seem to indicate HNCC tumors rather than specifically OSCC cancerous lesions.

Specific comments:

From table 1 it looks like the are three steps in the screening process. The second and third step are adequately explained of why rejections were made at that point, however, there seems to be no criteria that described exclusion at this point or the data that the Cohen kappa analysis was based on.

Line 156 indicates a Table 3, but there is no Table 3 in the manuscript.

The final sentence in the discussion the use of the word equivalent is confusing to me. Are the authors proposing that the amount of HPT hormones produced by the oral mucosa is equivalent to the endocrines produced by the actual HPT organs?

The authors indicate supplementary data, but these were not available for the review.

Author Response

Please find attached point-by-point response 

Reviewer 2 Report

This paper discusses the role of HPT hormones in the context of normal and cancerous oral tissues. In my opinion the article is well written the findings are well presented. Following is my comment on one of the sections.

1) Section 2.2.7. I'm not entirely sure of the point of this section with regards to the review presented in the manuscript. It offers no insight about the role of HPT and oral keratinocytes in the context of cancer progression nor have the authors elaborated more about this section in the discussion.

Author Response

(The authors gave the same response as above.)

Round 2

Reviewer 1 Report

The authors have addressed my concerns with sufficient clarity.